# Predicting Prognosis and Platinum Resistance in Ovarian Cancer: Role of Immunohistochemistry Biomarkers

**DOI:** 10.3390/ijms24031973

**Published:** 2023-01-19

**Authors:** Ghofraan Abdulsalam Atallah, Nirmala Chandralega Kampan, Kah Teik Chew, Norfilza Mohd Mokhtar, Reena Rahayu Md Zin, Mohamad Nasir bin Shafiee, Nor Haslinda binti Abd. Aziz

**Affiliations:** 1Department of Obstetrics and Gynaecology, Faculty of Medicine, University Kebangsaan Malaysia, Bandar Tun Razak 56000, Kuala Lumpur, Malaysia; 2Department of Physiology, Faculty of Medicine, University Kebangsaan Malaysia, Bandar Tun Razak 56000, Kuala Lumpur, Malaysia; 3Department of Pathology, Faculty of Medicine, University Kebangsaan Malaysia, Bandar Tun Razak 56000, Kuala Lumpur, Malaysia

**Keywords:** immunohistochemistry, biomarkers, platinum chemotherapy, chemotherapy resistance, disease relapse, ovarian cancer

## Abstract

Ovarian cancer is a lethal reproductive tumour affecting women worldwide. The advancement in presentation and occurrence of chemoresistance are the key factors for poor survival among ovarian cancer women. Surgical debulking was the mainstay of systemic treatment for ovarian cancer, which was followed by a successful start to platinum-based chemotherapy. However, most women develop platinum resistance and relapse within six months of receiving first-line treatment. Thus, there is a great need to identify biomarkers to predict platinum resistance before enrolment into chemotherapy, which would facilitate individualized targeted therapy for these subgroups of patients to ensure better survival and an improved quality of life and overall outcome. Harnessing the immune response through immunotherapy approaches has changed the treatment way for patients with cancer. The immune outline has emerged as a beneficial tool for recognizing predictive and prognostic biomarkers clinically. Studying the tumour microenvironment (TME) of ovarian cancer tissue may provide awareness of actionable targets for enhancing chemotherapy outcomes and quality of life. This review analyses the relevance of immunohistochemistry biomarkers as prognostic biomarkers in predicting chemotherapy resistance and improving the quality of life in ovarian cancer.

## 1. Introduction

Ovarian cancer is considered the fifth leading cause of cancer-related death in women. Worldwide, an estimated 313,959 new ovarian cancer cases and 207,252 ovarian cancer-related deaths occurred in 2020 [1]. Despite the improvement in treatment for ovarian cancer, survival trends remained poor due to chemoresistance and a lack of biomarkers to detect the disease early [2,3]. Hence, ovarian cancer is often diagnosed at an advanced stage, with most new cases spreading beyond the primary site. Over the last 30 years, mortality rates from ovarian cancer have remained poor [4], with patients with advanced disease (Stage III and IV) having a 10-year survival rate of 10–30%.

Epithelial ovarian cancer (EOC) originates from ovarian surface epithelium (mesothelium) and accounts for more than 85% of all ovarian tumours. The heterogeneity of EOC, which consists of several tumour subtypes with greatly divergent clinicopathologic characteristics and behaviour, poses a major challenge to understanding the pathophysiology of the disease. Various patient and tumour parameters, such as age, genetic makeup, and tumour traits including stage, grade, histologic subtype, and chemotherapy sensitivity, therefore, have an impact on the prognosis of ovarian cancer [5,6,7,8,9]. 

High-grade serous ovarian cancer (HGSOC) is the most common and aggressive form of EOC, which accounts for about ~70% of all cases and it is the leading cause of cancer-related death among all gynaecological cancers worldwide [10]. Less common types of epithelial ovarian cancers include: endometrioid carcinoma, which consists ofabout ~20% of EOC and occurs more commonly in women with endometriosis [11]; low-grade serous ovarian carcinoma (LGSOC), which is a slow-growing tumour that accounts for about 5% of EOC [12]; mucinous carcinoma, which is more distinct and tends to be large (around 8 inches or 20 cm) [13]; and ovarian clear cell carcinoma (OCCC) accounts for approximately 5% of all ovarian carcinomas and is characterized by a high recurrence rate [11] (Figure 1).

Optimal cytoreductive surgery and platinum-based chemotherapy using the combined carboplatin-paclitaxel regimen have been the standard treatments for EOC [14]. Despite an initial good response to first-line therapy, the development of chemotherapy-resistant and refractory diseases ensues. Therefore, the sensitivity of chemotherapy has decreased [5] with an increased relapse rate and, therefore, a decrease in long-term survival rate for ovarian cancer. It was shown that up to two-thirds of patients with advanced ovarian cancer experience cancer recurrence within 18 months from the time of diagnosis regardless of the first-line therapy [15]. Patients undergo platinum chemotherapy are classified into platinum-sensitive or platinum-resistant according to the time from the end of treatment to the recurrence of the disease (platinum-free interval). Platinum resistance, defined as disease recurrence within 6 months of completion of first-line platinum-based chemotherapy, occurs in approximately 25% of cases [16] and the median progression-free survival (PFS) is only 9–12 months on average [17]. Conversely, platinum-sensitive patients have a PFS for up to 24 months [18,19]. Currently, only paclitaxel, pegylated liposomal doxorubicin (PLD), and topotecan are approved by the US FDA to treat platinum-resistant ovarian cancer patients; however, the response rates are poor at 10% to 15% [5,15,18,19]. The taxane analogues, oral etoposide, pemetrexed, and bevacizumab are additional medications with some effectiveness in platinum-resistant ovarian cancer [5,15]. Unfortunately, second-line chemotherapy in patients with platinum-resistant ovarian cancer has not been found to be superior to current therapy in terms of progression-free survival or overall survival in randomised phase III trials [20].

The tumour microenvironment in ovarian cancer tissues is associated with altered protein expression patterns, making it conceivably a site of interest to decipher protein profile patterns and alteration in disease development and treatment intervention [5]. The discovery of molecular tumour traits linked to high-risk early-stage ovarian carcinomas would also enhance risk assessment, maybe have an impact on treatment selection, and direct the development of targeted therapies in the future.

Immunohistochemistry (IHC) is a very sensitive and unique technique used to determine tissue constituents (the antigens) with the employment of specific antibodies that can be visualised via a microscope. It has shown to be a potent technique for the identification and use of biomarkers, an example of this role includes the Human Epidermal Growth Factor Receptor 2 (HER2) expression in breast cancer and gastroesophageal adenocarcinoma, in addition to the expression of mismatch repair (MMR) proteins in patients with colorectal adenocarcinoma or endometrial carcinoma [21,22]. The immunohistochemical reactions have been used in different situations within the research or pathological context, the most important applications are: (1) histogenetic diagnosis [23]; (2) subtyping of neoplasia [24]; (3) characterisation of the primary site of malignant neoplasia’s [25]; (4) research for prognostic factors and therapeutic indications of some diseases [26]; and (5) discrimination of benign versus the malignant nature of certain cell proliferation [27]. Although IHC-based platforms for assessing the tumour immune milieu are easily implemented in clinical settings, several IHC-based biomarkers have struggled to achieve therapeutic relevance due to a lack of validation and inaccurate clinical outcome prediction [28]. 

This narrative review discusses the relevance of IHC-based biomarkers in predicting chemotherapy resistance and prognosis in ovarian cancer while also outlining the drawbacks of using IHC in clinical practice.

### Search Strategy of Review 

Keywords and terms of the major concepts for this review included: “Immunohistochemistry (IHC)”, Ovarian Cancer Subtypes”, “Ovarian Cancer Prognostic Value”, “Chemotherapy Resistance”, and “Ovarian Cancer Biomarkers”, which were developed and combined to form the search strategy. In the section entitled “Signature Alterations in Women with Chemotherapy Resistance” we included more specific keywords of ovarian cancer molecular pathways (e.g., “Tumor Mutation burden”, “DNA Repair Pathways”, and “Cell Cycle Related Genes”. The systematic search of Google Scholar, PubMed, Web of Science, and Sci-Hub databases were combined for relevant English-language publications from the time of their inception to October 2022. Results were merged using reference management software (Endnote X9; Thomson Reuters version 12.0.0.2401). The findings of relevant studies are summarized in Table 1 and Table 2.

## 2. Signature Alterations in Women with Chemotherapy Resistance

Chemotherapy resistance is linked to multiple mechanisms which include changes in the transport and cellular turnover of the drug as well as alterations in cytoplasmic defence systems and DNA repair mechanisms leading to the loss of treatment sensitivity [69]. The conventional chemotherapy drugs exert their function not only via cellular machinery that controls the cell cycle, but also through molecular pathways that mediate programmed cell death or apoptosis [70].

Various molecular markers including biomarkers derived from genomic abnormalities (such as gene mutations, copy number aberrations, and DNA methylation) have been associated with different types of EOC. Some of these biomarkers can predict disease prognosis and chemosensitivity. In addition, over the past ten years, numerous studies have found protein expression differences by examining a large number of ovarian tumour tissues [66,71,72,73].

Ovarian cancers are often linked to the genetic mutations of the BRCA genes and p53 mutations which occur in 50–80% of epithelial ovarian cancer, amplification, and overexpression of HER2/neugene and the AKT2 gene in about 10–20% of the high-grade serous carcinoma and inactivation of p16 gene recorded in 10–17% of the epithelial ovarian cancers [74,75]. Other suppressors and oncogenes, such as KRAS, *PTEN*, PIK*3CA*, *ARID1A*, *PPP2R1A* PTEN, BCL2, MYC, *BRAF*, *ERBB2*, *CTNNB1*, and TGF-β have been involved in the tumorigenesis of ovarian carcinoma and chemotherapy resistance [76,77,78]. While significant efforts have been made to understand the molecular processes underlying ovarian chemotherapy resistance, its pathogenesis and progression model remains unexplained.

The identification of molecular signatures is becoming more important for individualized targeted ovarian cancer treatment. From a therapeutic standpoint, the discovery of biomarkers has a significant role in predicting the results of chemotherapy treatment, which is essential in assisting clinicians in weighing the possibility of chemotherapy resistance and predicting the quality of life after chemotherapy [76].

## 3. Immunohistochemistry Biomarkers of Chemotherapy Resistance in Ovarian Cancer

Biomarkers are classified into two subtypes: prognostic and predictive markers. Prognostic indicators reveal disease outcome, while predictive markers refer to how well a patient responds to chemotherapy, which can be a therapeutic factor. To date, limited information is available for prognostic biomarkers associated with disease relapse and chemotherapy resistance in ovarian cancer. To improve the management of clinical outcomes; biomarkers must be specific and sensitive, and they should be convenient and inexpensive to be tested. Hence, studying the biomarkers within the tumour tissue microenvironment may yield better success for the potential discovery of relevant and critical biomarkers for prognostication and as therapeutic targets. Therefore, efforts are ongoing to identify IHC biomarkers as potential prognostic markers for ovarian cancer in the clinical setting. In this review, the findings from relevant studies on IHC biomarkers and their association with chemoresistance and prognosis are summarized in Table 1 for all types of EOC and Table 2 for subtypes HGSOC and OCCC.

### 3.1. Vascular Endothelial Growth Factor (VEGF)

It is a signal protein that many cells express that encourages the growth of blood vessels. By regulating tumour development through its support of tumour angiogenesis and ascites formation through its stimulation of vascular permeability, VEGF plays a significant role during ovarian cancer [79]. High levels of VEGF are linked with primary resistance to platinum-based chemotherapy, and the immunohistochemical level of VEGF expression is highly associated with platinum sensitivity and overall patient survival [80].

In another study, the author demonstrated that patients with platinum resistance had a higher proportion of VEGF levels compared to those in the platinum-sensitive group (86% vs. 2%) [81]. The median survival in the patient group with a high VEGF score was determined to be 11 months, compared to 32 months in the group with a low VEGF score, by the author [81]. Consequently, VEGF expression was inversely connected with overall survival (OS) and strongly correlated with platinum resistance EOC (*p* < 0.0001) [81]. When compared to the chemo-resistant patient group, the VEGF levels in the chemo-sensitive patients were significantly lower [81]. On the other hand, it was examined that the VEGF score in a multivariate regression model and found that the presence of VEGF in the tumour was a significant predictor of how well patients would respond to platinum-based chemotherapy. However, neither the stage nor the grade of the tumour, nor the patient’s age, were shown to be related to VEGF expression [82].

### 3.2. CD133

Often known as Prominin-1, the PROM1 gene on chromosome 5 encodes a five-transmembrane glycoprotein with a molecular weight of 97 kDa. CD133 is a putative marker for cancer stem cells in ovarian cancer, which have been known to predict resistance to chemotherapy [46,83]. According to earlier research, CD133-positive cells were more aggressive and tumorigenic in vitro and/or in vivo than their CD133-negative progeny and were more resistant to chemotherapeutic therapy [46,84]. Similarly, 31% of 400 ovarian cancer samples had CD133 expression, which, according to a log-rank test, was linked to both shorter OS times and shorter disease-free survival times (*p* = 0.007 and P0.001, respectively) [85]. The hypothesis that CD133 and cancer stem cells are related is supported by the finding that CD133 expression is a predictor of poor clinical outcomes for ovarian cancer patients [86].

### 3.3. P53

It is a tumour-suppressor gene located on the short arm of chromosome 17 [87], which regulates cell growth. One of the most often found genetic anomalies in human neoplasia is p53 gene mutations [88]. Numerous studies have examined p53 immunoreactivity in ovarian carcinomas, and the majority of these have found a significant percentage of positivity, particularly in serous tumours [10,89]. On the other hand, a recent investigation discovered functional p53 mutations in 50.8% of high-grade serous ovarian cancer (HGSOC) and 8.3% of low-grade ovarian cancer [90]. In addition, p53 was much more expressed in HGSOC than in low-grade patients (*p* = 0.005); whereas, only 18% (4 of 22) of low-grade cases showed 5+ staining and 64% (30 of 47) of HGSOC cases [90].

The time to progression and OS were considerably reduced in patients with p53 mutations compared to those with normal wild-type p53; however, since p53 mutations were discovered in 56 percent (99 of 178) of the epithelial ovarian tumours (*p* = 0.029 and *p* = 0.014). In 62 percent (110 of 178) of ovarian cancer tissues, p53 protein overexpression (>10 percent positively stained nuclei) was discovered [34]. In cases with p53 overexpression, the time to progression and OS were shorter (cut-point, 10%: *p* = 0.071 and *p* = 0.056) [34]. Individuals with p53 overexpression experienced resistance to adjuvant cisplatin or carboplatin treatment substantially more frequently (*p* = 0.001) than patients with normal p53 [34]. The effective induction of apoptosis by a functional p53 protein determines the sensitivity of tumour cells to various chemotherapeutic agents, and p53 loss can increase chemotherapy resistance [91,92,93]; however, opinions on the correlation between chemotherapy sensitivity and p53 status are still divided.

### 3.4. MIB-1/KI-67

The Ki-67 gene (10q25) is located on the long arm of human chromosome 10 [94]. The quantification of the reactive expression of Ki-67 antigen using an immunohistochemical tool has been demonstrated to provide an estimate of the tumour’s proliferative capacity and, therefore, has been widely utilized as a reliable prognostic marker in almost all types of cancers including those of the lymphatic system, lung, brain, breast, cervix, uterus, ovary, and soft tissue sarcoma [94,95]. The key feature of this biomarker is its expression is absent in the quiescent state of cells (G0) and expressed in all of the active cell cycle phases (G1, S, G2, and mitosis) in proliferating tissues [96]. The strongest areas of Ki-67 immunostaining are assessed, and all discernible nuclear staining—regardless of intensity—is considered positive immunoreactivity. Normally, immunostaining is restricted to the nucleus, and only mitosis is associated with cytoplasmic positivity. Using a high-power microscope objective, the percentage of positively stained cells (x400) expressed is defined as the Ki-67 labelling index (Ki-67 LI). Numerous malignancies, including ovarian cancer, have been demonstrated to have a poor prognosis with high Ki-67 LI [94,95].

Mindbomb E3 ubiquitin protein ligase1 (MIB-1 antibody) Immunohistochemistry (IHC) is a relatively new technique for determining the Proliferative Index (PI) of a neoplastic lesion [36]. The monoclonal mouse antibody MIB-1 is the standard for demonstrating PI in formalin-fixed, paraffin-embedded specimens for the Ki-67 antigen. It reacts with the nuclear protein Ki-67 antigen, which comes in two isoforms with molecular masses of 345 and 395 kDa [97]. Numerous studies have linked the presence of this IHC marker in EOC with other prognostic indicators such as histologic subtype, tumour grade, FIGO stage, treatment response, as well as with survival rates [94]. In addition, the mean MIB-1 index in HGSOC was found to be 55.4% compared with 23.0% in low-grade ovarian serous cancer [69]. Similarly, another study on the MIB-1 proliferative index (an independent predictor of lymph node metastasis) has shown a significantly lower value in low-grade ovarian cancer (16.3%) when compared with HGSOC (47.8%) [98].

Elevated Ki-67 LI was linked to high-grade tumours (69.9%), high-grade serous tumours (65.34%), and advanced FIGO staging (70.6%) in a study involving 202 women [36]. However, Ki-67 LI and CA 125 levels did not significantly correlate. The growth fraction of a tumour cell population can be determined with outstanding cost-effectiveness using the marker Ki-67. When paired with Ki-67 LI in the histopathology report, the histological grade and FIGO stage of EOC can aid in the diagnosis of subtype distinction, prognostication, determining whether adjuvant treatment is necessary, and survival analysis [97,99]. In a study involving seventy-three patients with EOC [100], Ki-67 expression in combination with survivin and Topoisomerase IIα was evaluated by immunohistochemistry on formalin-fixed, paraffin-embedded tissue sections in relation to the response to chemotherapy. Nuclear staining for all antibodies was scored in a three-layer system and staining >10% was accepted as expression. It was found that Ki-67 was related to poor OS (*p* = 0.005); however, there was no association between Ki-67 expression and histological subtype, stage or grade of ovarian cancer [98,100].

It has been well established that Ki-67 immunostaining of human tumours has a diagnostic and prognostic significance. The Ki-67 immunostaining has been carried out on a variety of histological and cytological specimens, including frozen sections, smears, cell suspensions, etc. However, the limitation of this epitope is that it does not survive conventional histopathological fixation, such as in formaldehyde or alcohol.

The regular inclusion of the Ki-67 LI/MIB-1 IHC marker as a diagnostic and prognostic feature in histopathology reports would pave the way for a better understanding of biological behaviour and the modification of treatment plans. The Ki-67 antigen/MIB-1 antibody reactive staining, therefore, can be employed as a diagnostic and predictive tool to direct the clinical care of ovarian cancer. Recent advances in methodological refinement have made Ki-67 antigen immunostaining a promising goal for PI [100].

### 3.5. The Mitotic Arrest Deficiency Protein 2 (MAD2)

It is essential to the operation of the Spindle Assembly Checkpoint (SAC), which mediates the attachment of spindle microtubules to kinetochores on chromosomes and the separation of chromosomes during mitosis. The expression was substantially higher in those with no recurrence compared to those with recurrence (24 vs. 17, *p* = 0.023) [101]. When the 41 cases were divided into low- and high-expression groups, the progression-free survival did not differ substantially between the two groups (*p* = 0.0685), but the low-expression group had a shorter OS compared to the high-expression group (*p* = 0.0188) [101]. This suggests that MAD2 expression levels can predict susceptibility to anticancer medications and the likelihood of recurrence. In addition, in a study of the immunohistochemical score of MAD2 protein was negatively correlated with progression-free survival of women with HGSOC (*p* = 0.0003), with a hazard ratio of 4.69. The reduced expression of MAD2 protein indicates a defective mitotic checkpoint, potentiates resistance to ovarian cancer cells that might eventually lead to a recurrence of the disease [102]. The paclitaxel-induced activation of mitotic cell death is the function of MAD2; however, when MAD2 is downregulated, the cellular response to paclitaxel is diminished.

### 3.6. Check Point Kinase 2 (Chk2)

It is a central key protein that mediates the response to genotoxic stress [103]. A positive response to platinum-based chemotherapy is associated with high Chk2 expression. In a study of immunohistochemistry involving 125 women with advanced stage HGSOC having a residual disease of less than 2 cm of cancer after surgery, it was shown that high expression of Chk2 was related to good response to platinum-based chemotherapy (OR = 0.132, *p* = 0.014) compared to those with low Chk2 expression in their pre-treated ovarian cancer tissues [38]. Similarly in a different study, Chk2 depletion reduced the platinum sensitivity of ovarian cancer cell lines, indicating that Chk2 should not be used as a therapeutic target in HGSOC patients because it eliminated the cisplatin-induced S-phase cell cycle arrest and increased long-term survival resistance to cisplatin [37].

### 3.7. Insulin-like Growth Factor 1 Receptor (IGF-1R)

It is a tyrosine kinase commonly found to be overexpressed in ovarian cancer women [81]. It is crucial for cell growth, differentiation, and death, and it may contribute to the development of cancer [104]. Numerous epidemiological studies looked at the connection between circulating IGF-1 levels and the risk of ovarian cancer, but no correlation was identified. An investigation of the IGF-1R levels’ prognostic significance in a small cohort of 19 women with HGSOC [48] found a significant rise in the expression of the IGF-1R transcript following six cycles of neoadjuvant chemotherapy (NACT) compared to chemo-naïve tumour tissues. The findings unveiled that the women with higher IGF-1R expression had prolonged disease-free survival (DFS: 26.7 months) compared to the ones with lower IGF-1R expression (DFS: 11.9 months). Increased plasma IGF-I levels were also more frequently found in well-differentiated epithelial ovarian carcinoma (*p* = 0.0047) [105].

### 3.8. Prostaglandin D2 (PGD2)

Prostaglandins are lipid-based arachidonic acid derivatives that regulate follicle-stimulating hormone (FSH)-mediated proliferation, differentiation, and steroidogenic activity in the normal ovary [106]. IHC evaluation of PGD2 is an independent marker of good prognosis in HGSOC. In a study of 114 HGSOC patients the IHC analysis revealed that a high expression of PGD2 correlated with improved disease-free survival (*p* = 0.009), the lack of relapse (*p* = 0.039), and platinum-based therapy sensitivity (*p* = 0.016). Therefore, this study concluded that the presence of PGD2 on ovarian tissue predicted a low risk of relapse when analyzed using multiple cox regression (hazard ratio, 0.37; *p* = 0.002), and therefore, was a good prognostic factor for women with HGSOC [49].

### 3.9. Endonuclease Non-Catalytic Subunit (ERCC1)

It is a protein critical in a nucleotide excision repair pathway. It was shown that NACT-treated HGSOC tissues showed a two-fold increase in ERCC1 expression compared to chemo-naïve HGSOC tissues (*p* < 0.0001) [50]. The neoadjuvant group with high ERCC1 had a mean overall survival of 141.6 months, which was noticeably longer than the ERCC1-absence group’s survival of 61 months (*p* = 0.028) [50]. A relationship between ERCC1 expression and tumour-infiltrating lymphocytes (TILs) is also suggested by this author, but further research is needed to confirm this. ERCC1, hence, can act as a potential biomarker that can predict platinum response and OS in ovarian cancer women undergoing NACT [107,108].

### 3.10. Notch Receptor 3

It is a bona fide oncogene, altered in approximately 20% of HGSOC women. It has a definite role in both the acquirement of chemoresistance and disease progression. Several studies have found Notch 3 as a significant prognostic factor in women with relapsed tumours [39,40,109]. In a study included 25 women with HGSOC have been followed up for a duration of 32 months. Out of all cases, nine (36%) clinically displayed tumour recurrence with acquired chemoresistance to first-line chemotherapy agents consisting of cisplatin and paclitaxel. The total cohort were divided into two: the higher-expressing group (n = 12) and the lower-expressing group (n = 13). The higher expression of Notch 3 (>2 fold) was significantly associated with chemo-resistant serous carcinomas compared to the low-expressing group (58.3% vs. 15.4%), suggestive of the role of Notch 3 as a possibly valuable predictive marker for chemoresistance [39]. In addition, it was highlighted that the possible association of Notch 3 and stage III/IV of ovarian adenocarcinoma with respect to poorer progression-free survival where 3 out of 5 women with a relapse (within 6 months post-first-line chemotherapy) had Notch 3 overexpression prior to chemotherapy treatment [40].

### 3.11. Glypican-3 (GPC3)

It is a heparan sulphate proteoglycan on the cell surface that attaches to the cell membrane through glycosylphosphatidylinositol anchors [61]. Its product is thought to interact with a variety of morphogenic or growth factors to control cellular development and apoptosis [110]. In a study of 213 cases with different subtypes of EOC, the GPC3 positive expression was demonstrated mainly (44%) in ovarian clear cell carcinoma (OCCC), and less in other subtypes including mucinous (4%), endometrioid (5%), and HGSOC (11%) [62]. Although GPC3 expression was significantly associated with poor overall survival in advanced (stage III/IV) OCCC, there was a negative correlation between GPC3 expression and clinicopathological aspects, such as tumour stage, lymph node spread peritoneal metastasis, and death rate [62]. Hence, GPC3 may be a potential marker for the advanced stages of OCCC [63].

### 3.12. Aldehyde Dehydrogenase (ALDH1)

Is a catalase that oxidizes aldehyde-containing molecules and ALDH; therefore, it plays an important role in cellular homeostasis. In cancer cells, it assists in both energy production via retinoic acid (RA) synthesis, and in the deactivation of drug molecules (by action on the aldehyde group) [51]. A growing body of research indicates that ALDH may not only be utilised as a marker for stem cells but may also control cellular processes such as self-renewal, growth, differentiation, and radiation and drug resistance. According to recent research, both healthy and cancer cells with high levels of ALDH1 activity have the capacity to serve as stem cells, as well as the capacity for self-renewal and stress resistance [52,111]. Many studies have shown that high expression of ALDH1 is found to be associated with increased capacity for sphere formation, tumorigenicity, and invasiveness [111,112].

There is strong evidence to support the idea that the specific isoform(s) of ALDH expressed in tissue determines its function as a stem cell marker. The cytosolic enzyme needed for RA production belongs to the ALDH1 family, and ALDH1A1 have received a lot of attention [111]. Both normal stem cells and stem-like cells that initiate tumours have their biological functions controlled by ALDH1A1, which encourages tumour growth and chemotherapy resistance [113].

It was reported that ovarian cancer can also be prognosticated poorly based on high ALDH1A1 expression in the immunohistochemistry of the ovarian tissue. Higher ALDH1 expression levels in ovary cancer cases were found to be associated with a worse prognosis in both serous (*p* = 0.006) and clear cell adenocarcinoma (*p* = 0.047) cases, according to immunohistochemical staining of a total of 123 EOC tissues [52]. ALDH1, hence, is a marker for ovarian cancer stem and the degree of ALDH1 expression may be a potential diagnostic for predicting a bad prognosis.

### 3.13. Homeobox A10 (HOXA10)

HOXA10 is a homeobox allotype gene in HOX family. In a single-center study conducted at Fudan Hospital, China, 29 women were evaluated for HOXA10 expression and correlation with survival [64]. According to Kaplan-Meier analysis, HOXA10 expression was negatively correlated with the 5-year survival rate, which was only 30% in the 20 women with positive HOXA10 expression and 55.6 percent in the nine women with absent HOXA10 expression [65]. This suggests that the Human homeobox gene A10 may be used as a prognostic factor in ovarian cancer and that HOXA10 could be a therapeutic target for this type of cancer.

### 3.14. AT-Rich Interaction Domain 1A (ARID1A)

BAF250a, the protein encoded by ARID1A, is one of the accessory subunits of the SWI–SNF complex chromatin remodelling complex which modulates the repression/de-repression of several promoters, and it acts as a tumour suppressor by nature [66]. Immunohistochemical analysis of 53 ovarian clear cell carcinoma (OCCC) patient samples was the first report demonstrating that low levels of ARID1A protein can serve as a marker of poor outcome in OCCC patients, whereby out of the 53 patients, eight with low ARID1A expression had shorter progression-free survival than those with high expression (*p* = 0.044, log-rank test) [67]. Furthermore, platinum-based chemotherapy significantly decreased OS (*p* = 0.03) and progression-free survival (*p* = 0.01) for nine patients with loss of ARID1A expression compared to those with positive ARID1A expression in a cohort of 60 patients with epithelial ovarian cancer diagnosed at stages I to IV [68]. This demonstrates that low levels of ARID1B protein can serve as a marker of poor outcomes in patients with ovarian cancer.

### 3.15. Hepatocyte Nuclear Factor-1β (HNF-1β)

HNF-1β is a homeodomain-containing transcription factor which binds to the same DNA sequence as homodimers or heterodimers. Numerous genes are known to be regulated by it, either directly or indirectly [114]. The majority of ovarian cancer clear cell carcinoma (OCCC) over-express HNF-1β; therefore, HNF1β over-expression is likely to be helpful for the diagnosis of OCCC. The presence of HNF1-binding sites at numerous OCCC-specific hypomethylated genes further supports this notion. In addition, a motif analysis found that HNF1 binding motifs are significantly enriched in genes that comprise the OCCC signature [115]. In two separate large cohort studies, one on an Australian population known as Australian ovarian cancer women (AOCS) and in another study on a Japanese population known as high-volume Japanese university clinical network (JIKEI), the over-expression of HNF-1β was found to be in OCCC subtypes and have been associated with significantly longer PFS (*p* = 0.01) and OS (*p* = 0.02) [54,116].

It was demonstrated that HNF-1β increases OCCC cell survival by enhancing Reactive Oxygen Species (ROS) resistance. Hence, HNF1β inhibition with some type of inhibitor, such as the microRNA mir-802, may yield a therapeutic effect by annulling ROS resistance [53]. The HNF1β-induced cell survival according to the author was shown to be glucose-dependent [53]. Therefore, glucose metabolism may be a therapeutic target in OCCC with high HNF-1β expression.

### 3.16. Cyclooxygenase-1 and 2 (COX-1 and COX- 2)

Cyclooxygenase plays important roles in catalysing rate-limiting reactions for prostaglandin and thromboxane synthesis and is often dysregulated in neoplastic tissues. COX-1 and COX-2 were often expressed in every type of epithelial ovarian cancer, suggesting that each may contribute to cancer development or progression [117]. The COX-2 expression was analysed by IHC in 87 women with ovarian cancer and Ferrandina et al., found that the percentage of positive COX-2 expression was significantly higher in non-responders than in patients responding to treatment (*p* = 0.043 and *p* = 0.0018, respectively) [41]. This is the first study to show a link between COX-2 and decreased chemotherapy susceptibility and poor outcomes in a large cohort of patients with primary advanced ovarian cancer who had the detectable disease at initial surgery [41]. Another study by Li et al., found that COX-1 protein was found over-expressed in 69.3% of the total 137 ovarian cancers [42]. COX-2 was present in 70.8% of all epithelial cancers’ subtypes, with 63.9% of the primary cancers and 81.5% of the metastatic cancers, positive for COX-2 [42]. The immunostaining for COX-2 was frequently found at the advancing margin of tumour invasion or in new metastatic, whereas the COX-1 protein overexpression was observed in ovarian surface epithelial cells, especially that of inclusion cysts [42]. Assessment of COX-2 status may provide additional information to identify patients with ovarian cancer who have a low chance of responding to chemotherapy and are potential candidates for personalized treatments.

### 3.17. Breast Cancer Gene 1 (BRAC1)

BRCA1 is reported to be downregulated in 15–72% of EOC cases [118,119]. Several retrospective studies report that BRCA1-mutated EOC women have a survival advantage attributed to the enhanced response to platinum chemotherapy [118,119,120]. In a study of 292 women with ovarian cancer, it was observed that 120 of the cases had ≤10% of the IHC nuclear staining of BRCA1 and 59% of women expressed >10% of BRCA1 staining which was classified as overexpression of BRCA1 [118]. Women with low BRAC1 expression had improved median OS (41.5 months) and PFS (16.3 months) compared to women with overexpression of BRCA1 in ovarian tumour tissues, where the median OS is 28.7 months and PFS is 13.4 months [119]. Similarly, women with absent/low levels of BRCA1 expression receiving platinum/taxane regimens had improved median OS (61.4 vs. 43.2 months) and PFS (23.2 vs. 18.2 months) when compared to other women with overexpression of BRCA1 [121]. This demonstrates that low BRCA1 expression can be used as a positive prognostic factor in ovarian cancer and to predict an enhanced response to platinum chemotherapy.

### 3.18. Programmed Cell Death Ligands (PD-L)

An immunoinhibitory receptor belonging to CD28/cytotoxic T lymphocyte antigen [122]. In a recent study looking at the PD-L1 and PD-L2 expression on ovarian tissues, it was reported that among 70 tissue samples obtained during primary surgery, the proportion of high expression (>20%) was 68.6% for PD-L1 and 37.1% for PD-L2 [44]. Women with high PD-L1 expression have shown a poor 5-year survival rate than those with low PD-L1 expression [123]. This suggests that the higher expression of PD-L1 on tumour cells leads to impaired antitumor immunity [124]. In another investigation, it was shown that PD-L1 on tumour cells directly suppresses antitumor CD8^+^ T cells [45]. The PD-L1 expression in ovarian tissues can be a marker for a poorer prognosis.

### 3.19. Forkhead Box Transcription Factor (FOXP3)

It is involved in the regulation and function of the immune system. FOXP3 plays a crucial role in the generation of immunosuppressive CD4^+^ CD25^+^ regulatory T cells (Tregs), which induce immune tolerance to antigens [125]. The presence of intraepithelial FOXP3+ cells was associated with increased disease-specific survival (*p* = 0.010) [55]. Moreover, ovarian cancers that were triply positive for intraepithelial CD4^+^, FOXP3^+^, and CD25^+^ cells showed a trend towards increased survival (*p* = 0.059) [55]. On the other hand, the quantitation of the FOXP3 expression in the patient subgroup (>81th percentile), is significantly associated with worse prognosis in terms of OS (27.8 versus 77.3 months, *p* = 0.0034) and progression-free survival (18 versus 57.5 months; *p* = 0.0041) when compared to those with less FOXP3 expression in women ovarian tissues [56]. High-expression levels of FOXP3 might represent a surrogate marker for an immunosuppressive milieu contributing to tumour immune escape [56,126].

### 3.20. Tumour Necrosis Factor Receptor 2 (TNFR2)

It has been found that a strong expression of TNFR2 on regulatory T cells (Tregs) in ovarian cancer tissue creates a potent immunosuppressive tumour microenvironment and is associated with poor clinical response [127]. A study looking at TNFR2 immunostaining intensity in ovarian tissues of 126 patients with ovarian cancer, found that the immunostaining intensity correlated with tumour stage (*p* < 0.001), stages I–II (30%) compared to stages III–IV (66%) [58]. Women with positive immunostaining to TNFR2 were found to have a significantly shorter mean survival time (*p* = 0.002). This study concluded that the tissue expression of TNFR2 in epithelial ovarian cancer correlated with the highest risk of cancer progression [58]. In another recent retrospective study, strong expression of TNFR2 was seen in the ovarian tissue of patients with chemo-naïve advanced ovarian cancers [57]. In this study, both the platinum-sensitive group (71.4%) and platinum resistance groups (81.8%), and the difference in the TNFR2 expression between the two groups were not statistically significant [57]. The PFS trend was longer in the weaker protein expression (5–50%) of TNFR2 compared to the stronger expression group (31 vs. 18 months) but this was not statistically significant. Women with TNFR2 over-expression had a longer median PFS interval of 540 days in the platinum-sensitive group, and a shorter interval of 90 days in the platinum-resistance group, *p* = 0.0001 [57]. The limitation of this study is its small sample size. The TNFR2 marker in IHC has the potential to be used as a positive indicator towards chemotherapy treatment and a larger prospective study may help to confirm its role.

### 3.21. Signal Transducer and Activator of Transcription 3 (STAT3)

The JAK/STAT3 activation pathway is thought to be crucial for a number of oncogenic activities, including tumour growth, differentiation, angiogenesis, and survival [128]. According to a number of studies, total STAT3 and phosphorylated STAT3 (p-STAT3) are overexpressed in a subgroup of chemotherapy-resistant ovarian cancer cell lines compared to their expression in the corresponding chemotherapy-sensitive cell lines [85,129,130,131]. It was shown that inhibiting STAT3 signalling could possibly abolish cisplatin resistance in ovarian cancer patients receiving chemotherapy because STAT3 is constantly active in cisplatin-resistant ovarian tumours.

A current systematic review and meta-analysis on 16 eligible studies involving 1747 ovarian cancer patients found that STAT3/p-STAT3 expression was upregulated in ovarian cancer samples compared to normal ovarian tissue, benign tumour, and borderline tumours (OR = 10.14, *p* < 0.00001; OR = 9.08, *p* < 0.00001; OR = 4.01, *p* < 0.00001) [59]. The STAT3/p-STAT3 overexpression also correlated with FIGO stages (I–II vs. III–IV) (OR = 0.36, *p* < 0.00001), tumour grades (G1 + G2 vs G3) (OR = 0.55; *p* = 0.001) and presence of lymph node metastasis (OR = 3.39; *p* < 0.00001). High STAT3/p-STAT3 expression was associated with shorter OS (HR = 1.67, *p* < 0.00001) and progression-free survival (PFS) (HR = 1.40, *p* = 0.007) [59]. This meta-analysis concluded that STAT3/p-STAT3 over-expression likely indicates a poor prognosis in ovarian cancer patients. Nevertheless, prospective studies are needed to confirm these associations. A recent retrospective study [57] found that the over-expression of STAT3 was seen in the chemo-naïve ovarian tissue of patients with advanced ovarian cancer (19/25, 76.0%) and in both platinum-sensitive (78.6%) and platinum-resistant (72.7%) groups. The PFS was longer trend in the weaker (5–50%) protein expression of STAT3 compared to the over-expressed (>50%) group (34 vs. 18 months) although this was not statistically significant due to the small sample size. Patients with STAT3 over-expression displayed a longer PFS of 120 days in the platinum-resistant group and a better PFS of 660 days in the platinum-sensitive group (*p* = 0.0001) [57]. To verify the results, a study with a bigger sample size will be needed.

## 4. Limitations Associated with the Immunohistochemistry Technique in the Clinical Settings

Despite the ubiquitous presence of IHC in research and diagnostic procedures, there are several limitations; Most notably, the lack of strict guidelines for staining often leads to conflicting results between different facilities using diverse procedures and various antibodies. In addition to consistency in antibody concentrations, many other components of IHC lack quality control. For example, whether an antibody binds to its target with appropriate sensitivity and specificity is not routinely tested. The lack of quality control procedures outside of the antibody itself may further contribute to the unreliability of staining. The sensitivity and specificity of the employed antibodies can be impacted by variations in tissue absorption time, slide thickness, and antigen retrieval. Therefore, to enable the widespread application of IHC-based biomarkers, thorough, standardised techniques are required. In addition, the optimisation of IHC is particularly important for newly discovered molecules or new antibodies. The specificity and sensitivity of the IHC need to be validated.

## 5. Future Directions

IHC platforms can be utilised in a clinical setting for disease diagnosis, early disease prognosis prediction, and early therapy response prediction. An automated image analysis platform is advised to be verified and deployed to reduce observer variability and be more reliable when quantifying biomarkers in patient samples to increase the reliability and reproducibility of IHC. However, because they rely heavily on user input to speed up the machine-learning process, these platforms are still only partially automated. Traditional IHC has been used to identify and study the biomarkers discussed in this review, but new technologies enable more in-depth analyses of molecular markers. For instance, technologies such as Vectra or AQUA that enable the inclusion of immunofluorescence enable the simultaneous investigation of many cellular phenotypes.

In our review, we found that the Ki-67 antigen/MIB-1 antibody immunostaining can be employed as a diagnostic and predictive tool to direct the clinical care of ovarian cancer. Glypican-3, ALDH1A1, TNFR2, STAT3, FOXP3, and TIM3 are increasingly recognised biomarkers to predict chemoresistance in women with ovarian cancer. In addition, HOXA10, HNF-1β, and ARID1A can be putative biomarkers with the potential to prognosticate the response to therapy. It was also shown that Chk2, PGD2, and NOTCH 3 are promising biomarkers for the prediction of chemoresistance in HGSOC women. On the other hand, MAD2, IGF1R, PDL-1, PDL-2, and ERCC1 are still under investigation, whereby clinical cohort studies with larger sample sizes and appropriate end-points are warranted to validate their potential. Evaluations of newer biomarkers are much needed to predict/evaluate the quality of life and overall survival in women with different subtypes of EOC.

Finally, although biomarkers can act as a stand-alone indicator, a single biomarker is often not sufficient to stratify women unequivocally and safely. Given that numerous biomarkers have been discovered using these methods, the integration of IHC with genomic and transcriptional techniques may help in the more accurate and predictive identification of biomarkers. IHC-based methods can therefore be reliable on their own, but by combining them with other tests or expanding them further to be more composite and quantitative in clinical research, we can hasten the development and confirmation of currently available biomarkers. Continuous understanding of these molecular mechanisms has the potential to pave the way for the creation of pharmaceutical treatments for cancer that are more precisely targeted.

## 6. Conclusions

Ovarian cancer is the most lethal gynaecological cancer; however, efforts are being made to improve the clinical outcome. IHC has become an essential tool for pathologists to elucidate the pathophysiology of the disease, both in routine practice and in research. IHC is also an important tool for validating biomarkers, which would eventually help in deciding the best treatment.

In this review, we attempted to provide a thorough search for the most effective IHC biomarkers associated with chemotherapy response and disease relapse as well as a brief highlight on the principles and practical tips for employing various biomarkers as a diagnostic and predictive tool for ovarian cancer women in tissue histopathology. In addition, this review is intended to be a useful platform for a future clinical study to implement post-operative biomarkers as a clinical prognostic tool and to customize cancer treatments.

## Figures and Tables

**Figure 1 ijms-24-01973-f001:**
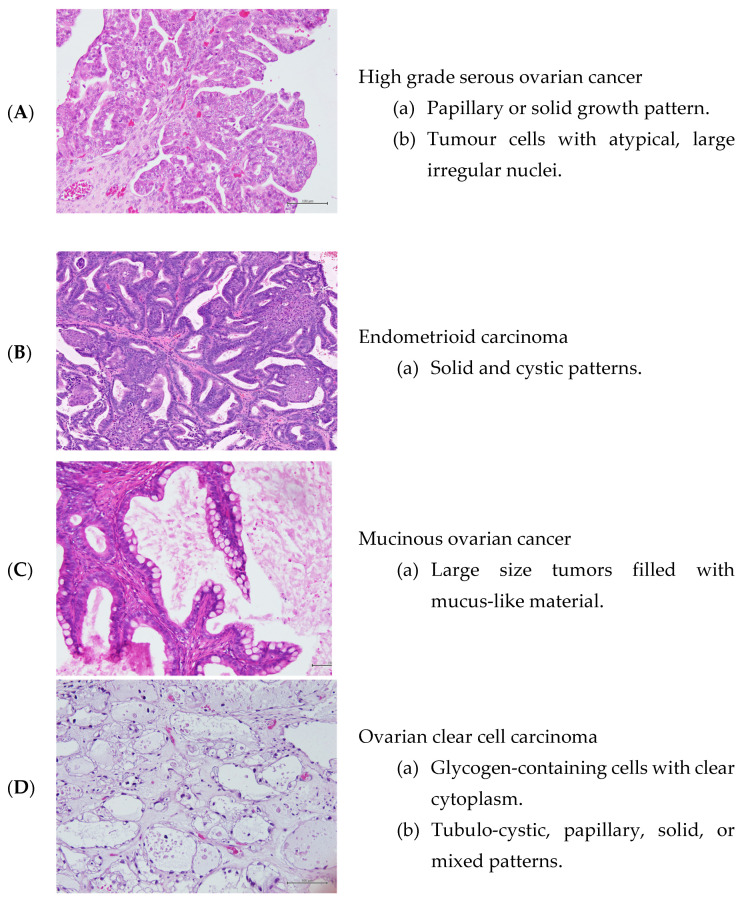
Hematoxylin and Eosin staining of the ovarian cancer tissues (under magnification of ×20).

**Table 1 ijms-24-01973-t001:** Summary of IHC biomarkers associated with chemoresistance and prognosis in EOC.

Biomarker	FIGO Stage	Study Design	Cut-Off Score of Expression	Outcome Measured	Clinical Indicator	Reference
VEGF	I–III	NR	10%	PFS, OS	Early detection of poor prognosis	[29,30,31]
P53	I–IV	NR	≥10%	OS	Poor survival	[32,33,34]
MIB1/KI67	1–IV	NR	≥50%	PFS	Poor survival	[33,35,36]
Chk2	I–IV	RC	≥50%	Chemotherapy response	Predicts good response to platinum-based chemotherapy	[37,38]
Notch3	I–II	RC	≥50%	Chemotherapy response, OS	Chemoresistance, poor OS	[39,40]
COX-1, COX- 2	NR	RC	>30%	OS	Poor OS indicator	[41,42]
BRCA1	NR	RC	≥10%	PFS, OS	Positive prognostic factor Predicts good response to platinum-based chemotherapy	[43]
PD-L1 & PD-L2	NR	RC	>20%	PFS, OS	Negative prognostic factor	[44,45]

Abbreviations: EOC, Epithelial Ovarian Cancer; FIGO, International Federation of Gynecology and Obstetrics; IHC, Immunohistochemistry; RC, retrospective cohort; OS, overall survival; PFS, progression-free survival; NR, not reported.

**Table 2 ijms-24-01973-t002:** Summary of IHC biomarkers associated with chemoresistance and prognosis in HGSOC and OCCC.

Biomarker	FIGO Stage	Study Design	Cut-Off Score of Expression	Outcome Measured	Clinical Indicator	Reference
Subtype—High-Grade Serous Ovarian Cancer (HGSOC)						
CD133	I–IV	RC	≥10%	OS	Early detection of poor survival	[46,47]
IGF-1R	I–IV	RC	≥5%	PFS, OS	Improves PFS and OS	[48]
PGD2	NR	RC	≥50%	PFS, response to chemotherapy	Predict good prognosis, PFS and chemotherapy sensitivity	[49]
ERCC1	NR	RC	>50%	PFS, OS	Predict longer PFS and OS	[50]
Aldh1a1	NR	RC	>20%	OS	Poor prognostic marker	[51,52]
HNF-1β	NR	RC	>30%	PFS	Chemotherapy resistance indicator	[53,54]
FOXP3	I–IV	RC	≥50%	DSS	Positive prognostic factors Negative prognostic factors, shorter OS and PFS	[55] [56]
TNFR2	NR	RC	>50%	PFS, OS	Shorter OS and PFS	[57,58]
STAT3	NR	RC	>50%	PFS	Shorter PFS	[57,59]
MAD2	I–III	RC	≤ 50%	PFS, OS	Resistance to paclitaxel, Shorter PFS	[60]
Subtype—Ovarian Clear Cell Carcinoma (OCCC)						
GPC3	I–II	RC	>15%	OS	Poor prognostic marker	[61,62,63]
Aldh1a1	NR	RC	>20%	OS	Poor prognostic marker	[51,52]
HOXA10	I–III	RC	≥50%	5-year survival	Poor indication of 5-year survival	[64,65,66]
ARID1A	NR	NR	0-≤40%	PFS, OS	Poor prognostic marker	[67,68]
HNF-1β	NR	RC	>30%	PFS	Chemotherapy resistance indicator	[53,54]

Abbreviations: HGSOC, High-Grade Serous Ovarian Cancer; OCCC, Ovarian Clear Cell Carcinoma; FIGO, International Federation of Gynecology and Obstetrics; IHC, Immunohistochemistry; RC, retrospective cohort; OS, overall survival; PFS, progression-free survival; DSS, Disease-specific survival; NR, not reported.

## Data Availability

No new data were created or analysed in this study. Data sharing is not applicable to this article.

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
