# Peer review of "Predicting Prognosis and Platinum Resistance in Ovarian Cancer: Role of Immunohistochemistry Biomarkers"

_ijms, 2023, doi:10.3390/ijms24031973_

Round 1

Reviewer 1 Report

Ovarian carcinoma comprehends at least five different malignant neoplasms that includes high- and low-grade serous carcinoma, endometrioid carcinoma, mucinous carcinoma, and clear cell carcinoma. These types have distinctive histological, molecular, and clinical features. New target therapies and tailored oncological approaches have demanded an integrated multidisciplinary advance in the setting of ovarian carcinoma. The need to implement a molecular-based classification in the worldwide diagnostic and therapeutic setting of ovarian cancer required a search for easy-to-use and cost-effective molecular-surrogate biomarkers, relying particularly on immunohistochemical analysis. Atallah et al., focuses on the role of immunohistochemistry (IHC) biomarkers as a molecular diagnostic approach to ovarian carcinomas.

The Review is not always clear and comprehensive. It would greatly benefit of some IHC examples. The description of the biomarkers is a “list” without internal connections and clear role in platinum resistance. References are poor and often not up-to-date. The authors should pay attention to the abbreviations and the nomenclature that should be consistent throughout the text.

Specific comments are found below:

-I would suggest a brief description of the two main subtypes of epithelial ovarian cancers.

-It would be helpful for the readers to include some examples of histological examinations.

-Table 1: It reads very busy. Perhaps, the authors can prepare two tables, one for each subtype (EOC vs HGSOC). This would allow to eliminate a little bit of text and simplify the Table.

-lines 100-104: this part in unclear. Please, modify it.

-lines 127-128: this part in unclear. Please, modify it.

-lines 172-173: this statement is not correct. A protein is not a tumor suppressor gene. Please, modify it.

-line 201: MIB1 is an antibody used in IHC and as a consequence cannot be overexpressed. This error is present several times and generates confusion. Please, modify it throughout the text.

-lines 204-205: this part in unclear. Please, modify it.

-line 211: what does it mean 16.3 vs 47.8%?

-lines 297-302: this part in unclear. Please, modify it.

-lines 310-311: this part in unclear. Please, modify it.

-lines 316-318: this part in unclear. Please, modify it.

-lines 354-357: this part in unclear. Please, modify it.

-lines 396-399: this part in unclear. Please, modify it.

Reviewer 2 Report

This review describes the role of immunohistochemical biomarkers in predicting platinum resistance in ovarian cancer. I have the following comments:

1.         What search strategies were used for this review?

2.         Which databases were searched?

3.         A systematic review should be performed when submitting case reports to a journal like this.

4.         What chemotherapy schemes are used in platinum resistant disease? This should be added in the introduction section.

5.         Please omit the sentence: ‘This review can significantly contribute to better management of ovarian cancer cases with the use of biomarkers at the diagnosis.’ At this point the authors should mention whether any biomarkers of platinum resistance have been prospectively validated.

Round 2

Reviewer 1 Report

The authors have addressed many of my concerns but I am sorry they have completely ignored the suggestion to add and/or up-date the reference list. I am sorry I have not found Figure 1. Did I miss it?

Reviewer 2 Report

All comments have been addressed and I believe the manuscript is suitable for publication